# An autoimmune disease risk variant: A *trans* master regulatory effect mediated by IRF1 under immune stimulation?

Margot Brandt[1,2,3], Sarah Kim-Hellmuth[1,2,4,5], Marcello Ziosi[1], Alper Gokden[1], Aaron Wolman[1], Nora Lam[1,6], Yocelyn Recinos[1,2,3], Zharko Daniloski[1,7], John A. Morris[1,7], Veit Hornung[8], Johannes Schumacher[9], Tuuli Lappalainen[1,2]*

**1** New York Genome Center, New York, New York, United States of America, **2** Department of Systems Biology, Columbia University, New York, New York, United States of America, **3** Integrated Program in Cellular, Molecular, and Biomedical Studies, Columbia University, New York, New York, United States of America, **4** Statistical Genetics, Max Planck Institute of Psychiatry, Munich, Germany, **5** Dr. von Hauner Children's Hospital, Department of Pediatrics, University Hospital LMU Munich, Munich, Germany, **6** Program of Pathobiology and Mechanisms of Disease, Columbia University, New York, New York, United States of America, **7** New York University, Department of Biology, New York, New York, United States of America, **8** Gene Center and Department of Biochemistry, Ludwig-Maximilians-Universität München, Munich, Germany, **9** Center for Human Genetics, University Hospital Marburg, Marburg, Germany

* tlappalainen@nygenome.org

**Data Availability Statement:** The RNA-sequencing fastq files and gene expression matrices are available through NCBI's gene expression omnibus under series accession number GSE145487.

## Abstract

Functional mechanisms remain unknown for most genetic loci associated to complex human traits and diseases. In this study, we first mapped *trans*-eQTLs in a data set of primary monocytes stimulated with LPS, and discovered that a risk variant for autoimmune disease, rs17622517 in an intron of *C5ORF56*, affects the expression of the transcription factor *IRF1* 20 kb away. The cis-regulatory effect specific to *IRF1* is active under early immune stimulus, with a large number of *trans*-eQTL effects across the genome under late LPS response. Using CRISPRi silencing, we showed that perturbation of the SNP locus downregulates *IRF1* and causes widespread transcriptional effects. Genome editing by CRISPR had suggestive recapitulation of the LPS-specific *trans*-eQTL signal and lent support for the rs17622517 site being functional. Our results suggest that this common genetic variant affects inter-individual response to immune stimuli via regulation of *IRF1*. For this autoimmune GWAS locus, our work provides evidence of the functional variant, demonstrates a condition-specific enhancer effect, identifies *IRF1* as the likely causal gene in *cis*, and indicates that overactivation of the downstream immune-related pathway may be the cellular mechanism increasing disease risk. This work not only provides rare experimental validation of a master-regulatory *trans*-eQTL, but also demonstrates the power of eQTL mapping to build mechanistic hypotheses amenable for experimental follow-up using the CRISPR toolkit.

**Funding:** This work was supported by National Institutes of Health grants R01MH106842 (T.L.), Roy and Diana Vagelos Precision Medicine Initiative Pilot Grant (T.L.), T32GM008224 (Y.R.), and Marie-Skłodowska Curie fellowship H2020 Grant 706636 (S.K-H.), American Heart Association Postdoctoral Fellowship 20POST35220040 (Z.D) and Banting Postdoctoral Fellowship from the Canadian Institutes of Health Research (J.A.M). The funders had no role in study design, data collection and analysis, decision to publish, or preparation of the manuscript.

**Competing interests:** I have read the journal's policy and the authors of this manuscript have the following competing interests: T.L. is an advisor to Variant Bio, Goldfinch Bio, and GSK and has stock in Variant Bio. Other authors have declared that no competing interests exist.

## Author summary

Although many genetic loci have been associated to disease, understanding how these variants impact molecular and cellular functions to impact disease risk have been challenging. Here, we first used blood cells from a large number of individuals and stimulated them in the laboratory with a proxy for bacterial infection. We identified that a genetic variant associated to autoimmune diseases also affects the expression of the nearby transcription factor *IRF1* gene in early immune response, followed by expression change of other genes in late immune response. We then studied this effect in cell lines, using the CRISPR approach to silence the activity of the genomic element of this variant and cause mutations at that position. We found evidence that this autoimmune disease -associated variant is located in a genomic regulatory element that responds to immune stimulus and affects expression of *IRF1* and a complex gene regulatory network. Thus, our characterization of genetic regulatory variation in the human population combined with experimental follow-up suggests a plausible, previously uncharacterized molecular mechanism that may underlie this genetic variant's effect on immune disease risk.

## Introduction

The discovery of tens of thousands of genetic loci associated to complex diseases and traits has introduced the challenge of characterizing the biological mechanisms that mediate these associations. Most GWAS loci include a large number of associated variants in linkage disequilibrium (LD) and are located in noncoding regions with likely gene regulatory functions. Therefore, it is typically unknown what is the causal variant, affected regulatory element, target gene in *cis*, and downstream molecular pathways that mediate genetic associations to complex disease phenotypes. Furthermore, the cellular context where these effects take place is often elusive, with increasing evidence that effects can be not only specific to cell types, but also cell states.

A key approach to answer these questions is expression quantitative trait locus (eQTL) analysis to discover associations between genetic variation and gene expression. Most widely applied in *cis*, eQTLs across diverse tissues have been shown to be strongly enriched for GWAS signals and have the ability to pinpoint potential target genes in GWAS loci [1]. Furthermore, response *cis*-eQTL mapping for immune cells with *in vitro* stimuli have provided powerful evidence of genetic regulatory effects that are not only specific to tissues but also to cell states, and disease associations that are driven by disrupted response to environmental stimuli [2–5]. *Trans*-eQTL mapping for distant, typically interchromosomal, genetic regulatory effects has the additional potential for elucidating regulatory pathways of the cell, and *trans*-eQTLs have been shown to be very strongly enriched for GWAS signals [6–9]. Unfortunately, robust discovery of *trans*-eQTLs has been challenging due to the large multiple testing burden, smaller effect sizes, and higher tissue-specificity as compared to *cis*-eQTLs [6]. While a large number of *trans*-eQTLs are mediated by a *cis*-eQTL, indicating potential biological mechanism, *trans*-eQTLs in humans rarely have strong master-regulatory effects on pathways of multiple genes [6,9–11].

Despite the power of eQTL characterization, the approach has limitations that open up possibilities for complementary experimental analysis. Association studies cannot provide definitive evidence for distinguishing causal regulatory variants from their LD proxies, creating the need for massively parallel screens and CRISPR assays to test whether specific genetic perturbations indeed affect gene regulation. Furthermore, experimental characterization and validation of eQTL associations has been sparse, which is a problem especially for *trans*-eQTLs that do not always replicate well between independent data sets [12]. Finally, with the functional

genomics toolkit now including eQTL mapping and diverse experimental approaches in cellular models, understanding the transferability of *ex vivo* eQTL results and cellular models is poorly known. Understanding these aspects is a particularly burning question for experimental follow-up of disease-associated loci.

In this study, we mapped *trans*-eQTLs in a monocyte data set with LPS stimulus time points, with *cis*-eQTL discovery reported in earlier work [3]. We discovered an inflammatory bowel disease (IBD) GWAS variant rs17622517 that affects the expression of the transcription factor *IRF1* in *cis* under early immune stimulus, and a large number of genes in *trans* under late stimulus. Using CRISPRi silencing and CRISPR genome editing, we show that the SNP locus indeed functions as an enhancer, identify the causal variant of this association, and demonstrate that the LPS-specific *trans*-eQTL signal can be recapitulated in a cellular model. Our results suggest that this common genetic variant affects inter-individual response to immune stimuli via regulation of *IRF1*, providing a strong hypothesis for the functional mechanism of the IBD risk association in this locus.

## Results

### eQTL discovery and characterization in cis and trans

We mapped response eQTLs in *cis* in primary monocytes under baseline and under the innate immune stimulus with LPS, using a data set of 134 donors with SNP genotyping and gene expression measured under baseline, and early (90 mins) and late (6h) LPS treatment that triggers the Toll-like receptor 4 (TLR4) pathway. The *cis*-eQTL discovery is described in [3,13], and here our focus was how early immune response *cis*-eQTLs may translate to later *trans*-eQTL effects. To this end, for the lead variants of the 126 *cis*-eQTLs at 90 min, we mapped *trans*-eQTLs for all the expressed probes at 6h. We discovered 47 probes with at least one *trans*-eQTL (*trans*-eGenes) at 5% false discovery rate (FDR) and 204 probes at 25% FDR, with 117 *cis*-eQTLs having at least one *trans* effect at 50% FDR (S1 Table and Fig 1A).

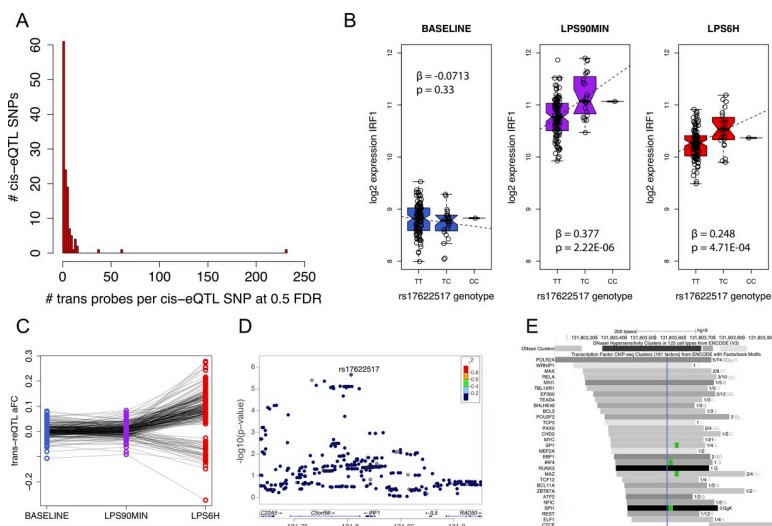

**Fig 1. eQTL analysis.** A) Histogram showing the number of trans-eGenes at 0.5 FDR per each tested cis-eQTL locus with 6h LPS stimulation. The rightmost bar is the IRF1 locus; B) the rs17622517 cis-eQTL for IRF1, and C) and its trans-eQTL effect sizes (allelic fold change) under the three conditions; D) the association landscape for the IRF1 cis-eQTL signal; E) overlap of rs17622517 (blue line) with ENCODE transcription factor ChIP-seq peaks (grey) and motifs (green). Intensity of grey scale is proportional to the intensity of the ChIP-seq peak.

One locus stood out by being associated with a larger number of *trans*-eGenes than any other: 12 at 5% FDR and 232 at 50% FDR (S1 Fig). Interestingly, the *trans*-eQTL is a cis-eQTL for the nearby *IRF1* only under early LPS response (Fig 1B), not for any other proximal gene (S2 Fig) and the *trans*-eGenes are enriched for *IRF1* target genes, defined as having an *IRF1* motif within 4 kb of their transcription start site (TSS) according to MSigDB (p = 1.8e-4). These *trans*-eQTLs are mostly active only at 6 hours after LPS stimulation (Fig 1C). This pattern supports a hypothesis where early LPS stimulation activates the *cis*-eQTL effect on *IRF1* first, which then affects the expression of downstream direct and indirect targets of *IRF1*. The alternative allele of the lead variant rs17622517 is associated with upregulation of *IRF1* upon early immune stimulus, and upregulation of 11 out of 12 of its most significant trans-eGenes as well (S3 Fig).

Next, we sought to further characterize the eQTL, its likely causal variant and its *cis*-regulatory mechanism. The lead variant rs17622517 is located 23 kb downstream of the *IRF1* TSS in an intron of the gene *C5ORF56* (Fig 1D). It has no LD-tagging variants at $r^2 > 0.2$ in our data set nor in 1000 Genomes data, nor structural variants that are not extremely rare in gnomAD. The other neighboring variants that are also significantly associated with *IRF1* in *cis* (S2 Table) represent an independent eQTL for *IRF1* (S4A Fig), with a lead variant rs147386065 that has a uniform effect under baseline and the two stimulus timepoints (S4B Fig). The trans-eQTL signal is significant only for rs17622517, but its effect sizes for the 50% FDR trans-eGenes are correlated to much weaker effect sizes from rs147386065 (S4C Fig), suggesting that both genetic effects on *IRF1* may contribute to downstream regulatory changes. For the lead eQTL, rs17622517 is a good candidate for the causal variant since it has no LD proxies, and its genomic context lends further support to this: In ENCODE data, the SNP locus overlaps open chromatin in many cell types, including monocytes [14], and a H3K27Ac peak in the GM12878 lymphoblastoid cell line suggests an active immune cell enhancer in this region. It also overlaps binding sites of numerous transcription factors in lymphoblastoid cell lines and is close to the motifs of many of them, including transcription factors activated in immune response, such as IRF4 and RELA, a subunit of NF-κB (Fig 1E).

Interestingly, this locus has a robust multi-trait genome-wide association signal for inflammatory bowel diseases (IBD; Crohn's disease, and ulcerative colitis) and ankylosing spondylitis [15–17]. At least three independent causal GWAS signals appear to exist in the locus, of which two match the two independent eQTLs in this locus and one of them is fine-mapped to rs17622517 [15,17]. The causal gene(s) in this locus have been unclear, with previous eQTL and other evidence pointing to several potential causal genes [15–17]. While *IRF1* is a plausible candidate as a key immune regulator with links to many ulcerative colitis-related genes [18–20], functional data implicating *IRF1* has been thus far lacking.

Our findings suggest that this GWAS association for IBD may be driven by a genetic effect that affects response to LPS via a cis-regulatory effect on *IRF1* followed by trans-regulatory effect on its downstream pathway. With a good candidate for a causal variant in a putative enhancer, we next sought to characterize and validate these effects experimentally using enhancer and promoter silencing by CRISPRi and genome editing by CRISPR.

## CRISPRi characterization of the enhancer function and downstream immune response

First, we had to select a cellular model. The monocyte-like THP1 cell line was used for some experiments, but since it is difficult to grow and edit, we chose to use as our primary model a modified HEK293 cell line HEK293/hTLR4A-MD2-CD14 (Invivogen) co-transfected with the human TLR4A, MD2 and CD14 genes, conferring NF-kB nuclear translocation in presence of

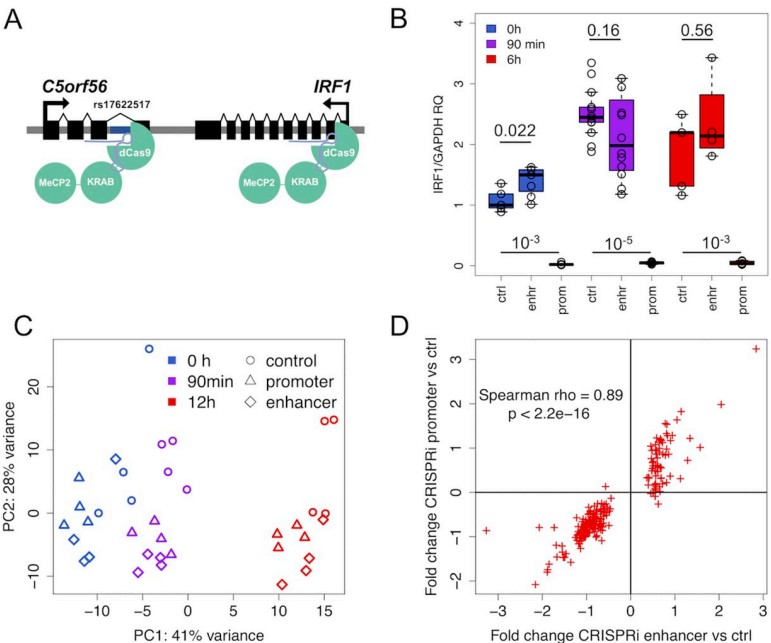

**Fig 2. CRISPRi silencing.** A) Illustration of CRISPRi silencing of the IRF1 promoter (right) and the putative enhancer locus at rs17622517 (left); B) qRT-PCR of IRF1 with promoter and enhancer silencing in THP1 cells; C) principal component analysis of gene expression under the different LPS and CRISPRi conditions; D) The correlation of expression fold changes for promoter and enhancer silencing at 12h LPS, shown for 225 genes that are significantly differentially expressed by enhancer silencing.

Gram-negative lipopolysaccharide (LPS) [21–23]. By RNA-sequencing after LPS stimulus under 4 concentrations and 7 timepoints we verified that the cell line responds to LPS stimulus and activates relevant pathways in a manner that is highly correlated to primary monocytes (S5 Fig). The transcriptional response appeared slightly delayed compared to primary monocytes, and thus we used 90 minutes and 12 hours post-LPS for early and late immune stimulus time points, respectively.

In order to study whether the rs17622517 locus indeed functions as an enhancer for *IRF1*, we first performed CRISPRi experiments in THP1 cell lines with the dCas9-KRAB-MeCP2 construct [24] (S6 Fig). We transfected the cells with gRNAs targeted to the variant locus to silence its activity (Fig 2A), the *IRF1* promoter 14 bp downstream of the *IRF1* TSS, and GFP as a control. We treated cells with LPS for 0h, 90m or 6h (with 4 to 14 replicates per condition), and performed qRT-PCR for *IRF1*. Its expression shows the expected upregulation with LPS in control samples (Wilcoxon p = 1.2x10$^{-4}$ at 90mins and p = 0.03 at 6h compared to 0h) and promoter silencing shows strong repression of *IRF1* (Fig 2B and S3 Table, Wilcoxon p <5x10$^{-3}$ for all time points), indicating a successful repressive effect of the promoter gRNA. Enhancer silencing leads to significant upregulation of *IRF1* at 0h LPS and suggestive downregulation of *IRF1* at 90m (Fig 2B and S3 Table, Wilcoxon p = 0.022 and p = 0.16, respectively). Although somewhat surprising that we see a significant effect at baseline rather than LPS stimulation, the subtle change in effect direction between baseline and 90m LPS of the genetic effect is consistent with the stimulation-specific eQTL results (Fig 1B). This supports the eQTL data suggesting that the locus is indeed a context-specific enhancer of *IRF1*.

Next, in order to study transcriptome-wide effect of this locus, we used HEK293/hTLR4A cell line that is easier to manipulate, and the same CRISPRi construct, gRNAs to target the enhancer and promoter loci, and LPS treatments for early and late immune response. We

performed RNA-sequencing on the resulting 36 samples with a median of 17 million reads, and analyzed gene expression patterns. Principal component analysis showed that samples cluster both by LPS condition and gRNA (Fig 2C), indicating that both promoter and enhancer silencing have strong effects on the transcriptome. We observe again an upregulation of *IRF1* expression with LPS (Wilcoxon p = 0.024 at 90mins and p = 0.038 at 12h compared to 0h in CRISPRi controls with GFP gRNA, S8 Fig). The fold change of 1.75 is comparable to that observed in the original primary monocyte data (1.23, Wilcoxon p = p$<2.2^{-16}$, S7 Fig). Promoter silencing shows strong repression of *IRF1* (Wilcoxon p = $7.4 \times 10^{-7}$ across time points), but the effect of the enhancer silencing is not significantly different from the controls for *IRF1* (Wilcoxon p = 0.8) or for other genes in the locus (P > 0.1) (S8 Fig). However, encouraged by the results in THP1 cells and the principal component analysis indicating a transcriptome-wide effect of the enhancer silencing, we hypothesized that we may be underpowered to detect its effect on *IRF1* in *cis* in this experiment, but might capture its *trans* effects by counteracting the noise present in an individual genes by looking at many genes at once.

Thus, we performed transcriptome-wide differential expression analysis to detect the transcriptional effect of LPS timepoints, gRNAs and the interaction of these. First, we verified a robust LPS response in the CRISPRi controls, with 30 differentially expressed genes at 90 minutes (FDR 5%) and 476 genes at 12h compared to the 0h samples (S4 Table), with both gene sets being enriched for GO categories related to inflammatory response (S5 Table). Next, analyzing the promoter and enhancer silencing effect compared to controls at 12h, we detected 1016 differentially expressed genes for the promoter and 225 genes for the enhancer silencing (S4 Table), both having an enrichment of many cellular processes including pathways related to immune response, LPS binding and TLR4 activation (S5 Table). The differential expression of promoter and enhancer have a strong positive correlation (rho = 0.89, p<2.2e-16, Fig 2D), which suggests that the impact of enhancer may be mediated by downregulation of *IRF1*. The enhancer silencing effect was stronger under LPS stimulus than under baseline (S9 Fig), suggesting that the enhancer is activated under immune response. Neither enhancer nor promoter differentially expressed gene lists were significantly enriched for msigdb IRF1 targets (Fisher test p = 1 and 0.245, respectively), possibly because differentially expressed genes are expected to include direct and indirect targets and downstream pathway effects of IRF1. Altogether, these results show that repression of the enhancer affects the transcriptome, indicating that it is an active regulatory region. Together with the qRT-PCR results showing that this enhancer affects *IRF1* and the transcriptome response being similar to the *IRF1* promoter silencing, our results indicate that the rs17622517 locus is an enhancer of *IRF1*.

## Detection of variant effects on gene expression by genome editing

Enhancer silencing does not necessarily reflect the effect that a genetic variant has on gene expression in *cis* and *trans* nor provide information whether a specific site has regulatory function. Thus, in order to test whether rs17622517 is indeed a functional eQTL variant and characterize its effect, we used CRISPR/Cas9 genome editing to introduce indels at the variant's genomic locus in HEK293/TLR4 cells; SNP editing in the locus with homology-directed repair had too low efficiency to be applicable. From a population of edited cells with 50% non-homologous end joining (NHEJ) based on targeted sequencing, we isolated and grew monoclonal cell lines, and genotyped them by sequencing. We focused on 1–20 bp deletions, allowing a mix of different edited alleles in a given clone, and because the HEK293 cell line is triploid at this locus, clones were considered heterozygous if they have one or two edited alleles. Fifteen clones with initially promising genotypes were further analyzed by long-range PCR to exclude one clone with a large rearrangement at the locus [25] (S10A Fig) and one heterozygous clone was

excluded due to a 50/50 allelic ratio suggesting that one allele had been lost. The 13 clones selected for functional follow-up include five homozygous reference (wild type) clones, three heterozygous clones, and five homozygous deletion clones (S10B Fig). Each of the 13 clones was exposed to LPS as above with 0h, 90 min, and 12h LPS treatment, after which RNA was extracted and sequenced, and gene expression was quantified. This was done in two replicates in order to reduce any potential noise especially in the stimulus step, and the reads from the replicates were combined for each sample, resulting in a median of 31 million reads per sample (S11 Fig).

The samples cluster strongly by LPS treatment in principal component analysis, as expected (Fig 3A and 3B). Differential expression analysis (S6 Table) confirmed the widespread LPS stimulus effect on gene expression of relevant pathways (S7 Table). *IRF1* expression is induced upon stimulation at 90m and 12h compared to the controls, but it is not significantly differentially expressed between genotypes (log2 fold change 0.37, p = 0.29 heterozygous vs wild-type, log2 fold change -0.03, p = 0.93 homozygous vs wild-type, log2 fold change -0.40, p = 0.26 homozygous vs heterozygous, Fig 3C). However, again hypothesizing that genetic perturbations at this locus could have downstream effects that we are underpowered to see for *IRF1*, we next analyzed transcriptome-wide differential gene expression. We used an interaction model between genotype and condition to see if clone genotypes differ in their transcriptional response to LPS stimulation, discovering only up to 37 differentially expressed genes (5% FDR, S6 Table) with no apparent link to IRF1 targets. Testing differential expression between

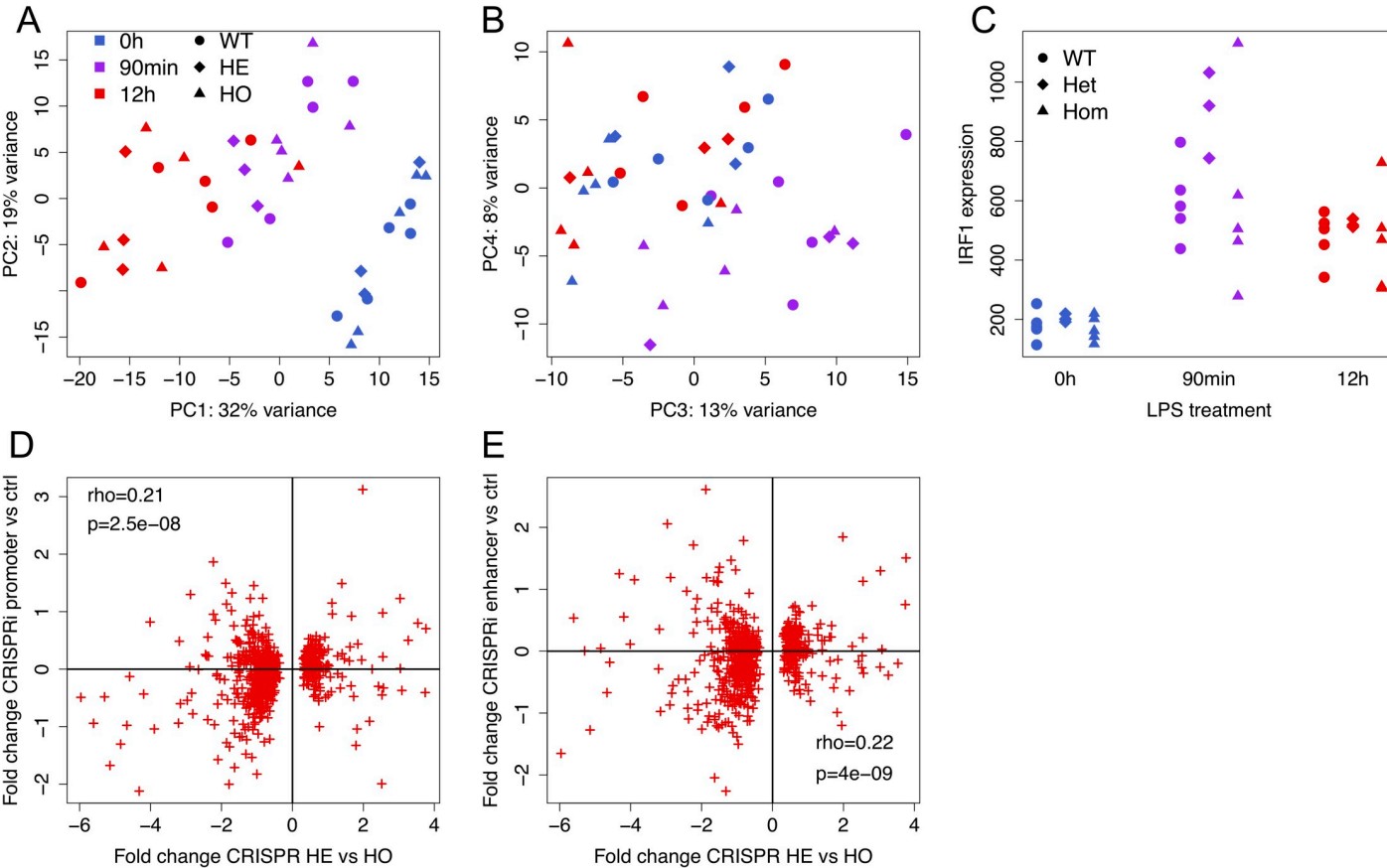

**Fig 3. Genome editing with CRISPR.** A-B) principal component analysis of gene expression for the different genotypes and LPS conditions, with principal components 1 and 2 in (A) and 3 and 4 in (B); C) IRF1 expression (gene counts normalized to sample read depth) for the different genotypes and conditions; D-E) Correlation of gene expression fold changes at 12h LPS condition for genes that are differentially expressed in heterozygous lines compared to homozygous lines at 0.25 FDR (x-axis), versus expression fold change for promoter (D) and enhancer (E) silencing (y-axis).

genotypes specifically at 12h, we saw the highest number of differentially expressed genes in comparison of homozygous and heterozygous clones (762 differentially expressed genes at 25% FDR, < 10 for other genotype comparisons). This discordance in numbers may be due to different effects of indels of slightly varying size and the cell lines potentially having other on-target or off-target mutations not detected by our genotyping assays (S10 Fig). Nevertheless, hypothesizing that these 762 genes may best reflect the impact of genomic disruption in this locus, we compared the gene expression response to that in the CRISPRi silencing experiment. We discovered a significant correlation of the fold changes with those from both promoter (rho = 0.21, p = 2.52e-8) and enhancer (rho = 0.22, p = 4.31e-9) silencing (Fig 3D and 3E). Correlations for the top 100 FC genes in other CRISPR genotype comparisons are shown in S12 Fig. These results suggest that disruption of genomic sequence at the rs17622517 locus can have functional effects that are partially similar to downregulation of *IRF1*, even though we are underpowered to detect effects on individual genes in *cis* and largely also in *trans*.

## Trans-eQTL effects in CRISPRi and CRISPR models

Having suggestive experimental evidence that rs17622517 is a functional eQTL variant and located in an enhancer that affects transcriptional immune response via *IRF1*, we sought to analyze whether the observed *trans*-eQTL effects in primary monocytes can be recapitulated in this experimental model. Thus, we analyzed trans-eGenes at a relaxed 50% FDR, and correlated their trans-eQTL effect sizes with expression fold change in CRISPRi and CRISPR experiments, all under the late LPS treatment. We observed a significant correlation of *trans*-eQTL effects with differential expression in the CRISPR cell lines for wild type/heterozygote (rho = 0.32, p = 3.7e-5, Fig 4A) and heterozygote/homozygote (rho = -0.27, p = 5.2e-4, S13B

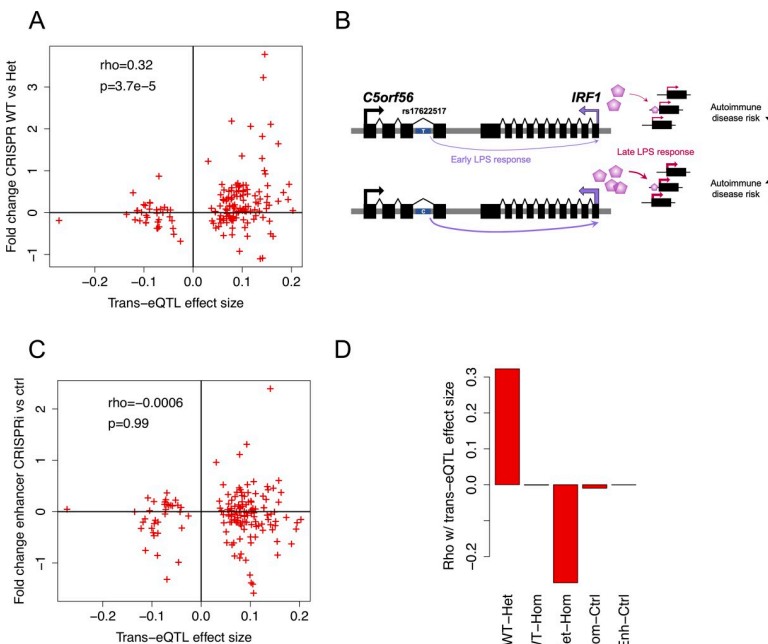

**Fig 4. Trans-eQTL effects captured by experimental data.** A) Correlation of trans-eQTL effect size with differential expression in CRISPR-edited heterozygous clones, B) Summary of the model supported by data of this study, with the rs17622517 C allele increasing the expression of IRF1 in cis under early immune stimulation, with downstream effects in trans under late immune stimulation. The C allele is also a risk allele for several autoimmune traits. C) Correlation of trans-eQTL effect size with differential expression after silencing of the enhancer locus with CRISPRi, and D) summarized for all the experimental comparisons.

Fig) comparisons. The opposing direction of the correlations might be explained by the slight pattern of IRF1 expression of the genotype classes (Fig 3C), with the heterozygous clones having upregulation and homozygotes having downregulation of IRF1, compared to wild type at 90 minutes. Altogether, these correlations suggest that the trans-eQTL is a true association that can be captured in our cellular model (Fig 4B) despite the differences between primary monocytes and HEK/TLR4 cells and the SNP versus CRISPR-induced indels. We observed no correlation of the trans-eQTL effects with CRISPRi enhancer or promoter silencing (Figs 4C, 4D and S13C). This suggests that while CRISPRi can be very powerful for demonstrating enhancer function, it may be suboptimal for recapitulating eQTL effects driven by genetic variants.

## Discussion

In this study, we discovered a condition-specific monocyte eQTL that affects the expression of *IRF1* in *cis* and many genes in its pathway in *trans*, so that alternative allele carriers have an increased upregulation of *IRF1* under immune stimulus. This is a plausible molecular mechanism for the inflammatory bowel disease association in this locus, with genetic predisposition for overactivation of *IRF1* immune response (by the rs17622517 C allele) possibly contributing to autoimmune disease risk. This is further strengthened by two independent eQTLs for *IRF1* having a GWAS signal for autoimmune traits, indicating convergence of genetic effects on autoimmune traits being mediated by *IRF1* expression.

We took a CRISPR-based approach to characterize this association in detail in a cellular model. Using a combination of enhancer and promoter silencing with CRISPRi and genome editing, we showed that the locus at rs17622517 is functional, and likely affects the activity of an immune stimulus -specific enhancer for *IRF1*, located in an intron of *C5ORF56*. The effect on the expression of downstream genes in *trans* suggests recapitulation of the trans-eQTL association and shows how a genetic effect on a transcription factor such as *IRF1* can affect entire downstream pathways.

Our work highlights both the power and challenges of CRISPR-based experimental analysis of functional effects of genetic variant. The eQTL mapping in primary monocytes under different treatments allowed us to build strong hypotheses of the mechanism that made the experimental follow-up practically feasible, demonstrating how observational functional genetics data helps to guide experimental work. Even though our modified HEK293 cell line is an imperfect model for events observed in primary monocytes, we were able to capture trans-eQTL effects in our cellular system. However, these effects do not fully recapitulate the *trans*-eQTL signal, and much more data will be needed to build a generalizable understanding of how well functional genetic effects can be extrapolated between cells in their physiological environment and *in vitro*.

Our results also highlight the limitations of CRISPR genome editing for functional follow-up: HDR efficiency is too low in many loci, including ours, for feasible analysis of specific SNPs, and interpretation of effects of different indels can be difficult. While editing the exact variant into a cell line may reflect directionally the same genetic effect as seen in the population, a deletion of a locus could feasibly have an effect in the same direction, opposite direction or not at all. This depends upon the specific mechanism of the genetic effect. For example, if a variant increases the affinity of a DNA binding protein by modifying its binding motif, that effect may not be replicated by a simple deletion of the variant locus. However, functional impact of indels does provide evidence that the specific SNP locus has a regulatory effect rather than being a nonfunctional LD proxy of another causal variant. Perhaps most importantly, we observed considerable variation between the different clones, possibly due to undesired and undetected genetic or epigenetic differences between the clones. This makes well-powered and

robust analysis difficult without a very large number of clones from each genotype. A key approach in this study to mitigate the limitations of genome editing was to complement it with CRISPRi silencing. Furthermore, while both CRISPR and CRISPRi analyses had limited power to detect robust effects on *IRF1* expression in *cis*, RNA-sequencing allowed us to leverage the power of the entire transcriptome, substantially improving our sensitivity to detect subtle regulatory effects.

The genetic eQTL effect on *IRF1* and downstream gene expression in this locus manifests only under immune stimulus, and thus an individual's transcriptional response to immune stimulus partially depends on the rs17622517 genotype. Such gene-environment interactions are common in *cis* [2–5,26], but few condition-specific master *trans*-regulatory variants such as the one characterized here have been well characterized before [2,27]. The association of this locus to inflammatory bowel disease, the known link between the IRF1 pathway and IBD [15–17], and the link between *IRF1* variants and eczema [28] and asthma [29,30] suggest that genetic *IRF1* dysregulation affects disease risk. While it is likely that rs17622517 is driving both the autoimmune disease signal and the immune response eQTL due to limited linkage disequilibrium with surrounding variants, the independent overlapping eQTL makes colocalization analysis of the locus difficult. Additionally, the specific mechanistic link between the transcriptional IRF1 response described here and autoimmune disease is unknown. However, *IRF1* is known to induce expression of type-I interferons [31] and pro-inflammatory cytokines, which can trigger an inflammatory response in a broad range of cell types [32]. Although this is a likely mechanism for *IRF1*'s role in autoimmune disease, further *in vitro* and/or *in vivo* experiments would be required to elucidate this disease mechanism and is a potential topic to be explored in future research. The role of *IRF1* in TLR4 signaling in response to LPS has not been well characterized [3,33], and our findings indicate that *IRF1* plays a role in LPS response in the innate immune system. However, IRF1 is expressed in many cell types and involved in other immune response pathways. Therefore, the effects described here and their relation to disease mechanism are not necessarily specific to LPS stimulation in monocytes, but could potentially be active in a wide range of immune cells and stimuli.

Altogether, our integrated analysis of genetic effects on gene expression in primary cells and in experimental models provides evidence that within the autoimmune disease risk locus near IRF1, rs17622517 is a functional variant that affects *IRF1* expression in monocytes under LPS stimulation and leads to overactivation of its downstream regulatory pathway.

## Methods

### eQTL discovery

The *IRF1 cis* response eQTLs (reQTLs) were discovered in a previous immune-response eQTL study[3]. 134 donors of German descent were included in the eQTL study. Individuals were genotyped using Illumina's HumanOmniExpress BeadChips comprising 730,525 SNPs, 579,090 of which passed QC and were used for analysis. Genotypes were phased with SHAPEIT2 and imputed with IMPUTE2 in 5 Mb chunks against the 1000 genomes phase 1 v3 reference panel. Primary monocytes were isolated from 134 donors and treated with LPS, MDP, IVT RNA, or no treatment. To assess early and late immune response, RNA expression was measured with a microarray chip after 90 min and 6 h. To detect a significant *cis*-reQTL, the effect size of association between genotype and expression of genes were compared between untreated and treated samples. rs17622517 was found to be a significant *cis*-reQTL variant for *IRF1* at 90m after LPS treatment, i.e. under early immune stimulus. To explore the possibility of another genetic variant driving the association, 1000G CEU data was analyzed to see if any variants were in linkage disequilibrium >0.2 with rs17622517. Additionally, structural variants

from Gnomad within 50Kb of rs17622517 were downloaded to see if any common structural variants were in proximity.

*Trans*-eQTL discovery was done with the MatrixEQTL R package, performing a linear regression between the genotype of the top variant for each of the 126 significant *cis*-reQTL at LPS 90 min and all expressed probes in monocytes at 6 hours (22,657). Benjamini-Hochberg correction was performed across p-values for all variants and probes. A significance threshold of FDR < 0.5 was used for downstream analyses. Since rs17622517 was found to be associated with many genes in *trans* at 6 hours, in addition to the *cis* association with *IRF1* and GWAS association for autoimmune diseases, it was selected for further follow up.

Effect size of trans-eQTL associations was measured as allelic fold change calculated from linear regression summary statistics [34]. When multiple expression probes measure the same gene, their mean effect size was used. Effect sizes from probes for the same gene were highly correlated (rho = 0.81, p = 2.5e-5).

## Enrichment of IRF1 targets

*IRF1* targets were obtained from the molecular signatures database (MSigDB) gene set *IRF1-01*, which comprises genes which contain at least one *IRF1* motif in the 4kb upstream and downstream from their transcription start site [35]. *Trans*-eGenes from the rs17622517 *trans*-eQTL (FDR < 0.5) and CRISPRi differentially expressed gene set (FDR < 0.05) were tested for enrichment in *IRF1* target genes using a Fisher's exact test comparing the number of *IRF1* target genes in these sets compared to the background of all expressed genes. The background set of genes was determined from the eQTL analysis as the probes with a p of detection < 0.01 in at least 10 samples across all conditions.

## Generation of THP1 CRISPRi cell line

Lentiviral particles were produced by transfecting HEK293FT (ThermoFisher) plated in 6-well plates at 80–90% confluency with 100uL Opti-MEM (ThermoFisher) pre-mixed with 0.55ug pMD2.G envelope plasmid (Addgene, 12259), 0.8ug psPAX2 packaging plasmid (Addgene, 12260), 1ug of transfer plasmid and 5.5ng of Polyethylenimine linear MW2000 (PEI) (Polyscience). Plasmids were mixed and incubated for 10 min, then the solution containing lentiviral particles was added to HEK293FT cells. We used two transfer plasmids to generate lentiviruses: KRAB-dCas9-MeCP2 (courtesy of the Sanjana lab). After 72 hours of incubation the media containing the lentivirus was harvested and collected and eventually concentrated with Lentivirus Precipitation Solution (Alstem Cell Advancem). THP1 cells were transduced at different virus concentrations in presence of 0.8ug/ml of Polybrene (Santa Cruz Biotechnology sc-134220). THP1 KRAB-dCas9-MeCP2 transduced cells were selected adding 10ug/ml of Blasticidin S.HCl (A.G.Scientific, B-1247) to the medium. Monoclonal cell lines where further selected and kept under selection at the same concentration of Blasticidin. sgRNA oligos were annealed and individually cloned by BsmBI digestion in pLentiGuidePuro vector (Addgene 52963) and verified by Sanger sequencing.

The following sgRNA sequences were inserted:

EGFP non targeting control (NT-Control): GGTGGTGCAGATGAACTTCA;

IRF1 promoter: GTCTTGCCTCGACTAAGGAG;

IRF1 enhancer: TTCTCTGTAGCCCTTGTATT.

Lentiviral production for the sgRNA lines was executed as above and THP1 KRAB-dCas9-MeCP2 monoclonal cell lines where transduced and selected in presence of blasticidin (10ug/ml) and puromycin (1ug/ml).

## Western blot to confirm THP1 cell lines

THP1 KRAB-dCas9_MeCP2 polyclonal and monoclonal lines were tested for the correct presence of the protein by Western Blot. 10ug of protein for each sample were denatured at 95C for 5 minutes in laemmli sample solution (BioRad 1610747). Protein was separated on 4–15% Mini-PROTEAN TGX Precast Protein Gels (Bio-Rad 451085DC). Protein was transferred on .45um Nitrocellulose membrane (Bio-Rad 1620117) and blocked in 5% milk in PBST for 1 hour. HA-tag present on the construct was detected by HA-Tag (6E2) primary antibody (Cell Signaling; Mouse mAb #2367; 1:2000) at expected size above 250kD. As an internal control, a primary antibody against GAPDH (Cell Signaling; Rabbit mAb #2118; 1:10000) was used. For detection, secondary Anti-Mouse IgG Polyclonal Antibody (IRDye 800CW; 1:10000) and Anti-Rabbit (IRDye 680RD; 1:10000) were used.

## Expression of IRF1 by quantitative RT PCR

Cells were grown in absence of LPS (time 0) or in presence of 200ng/ml of ultrapure LPS from Escherichia coli (invivogen) for 90 minutes and 6 hours. Total RNA from THP1 cells was extracted using TRIzol (Invitrogen) and Direct-zol RNA MicroPrep (Zymo Research R2062) and quantified by NanoDrop. RNA was stored at −80°C. Samples were digested with ezDNase before reverse trascription (RT) executed with 0.5 μg of RNA using SuperScript IV VILO (Sigma Aldrich 11766050).

To determine IRF1 gene expression level, quantitative RT-PCR was performed using Taq-Man Assays using IRF1 probe from Thermo Fisher Scientific (Hs00971965_m1, FAM). The expression of IRF1 was quantified by $-\Delta\Delta C\,T$ method and normalized to the multiplexed expression of GAPDH (Hs02786624_g1 VIC-MGB_PL). Four independent experiments were executed and run in technical triplicates in two quantitative PCR. Samples with GAPDH Ct above a threshold of 27 were excluded from the analysis as indication of low cDNA input or inconsistent PCR measurement and biological replicates were excluded if they had less than two technical replicates. Biological replicates included in the final analysis consisted of 6 0h control, 8 0h promoter, 7 0h enhancer, 14 90m control, 7 90m promoter, 12 90m enhancer, 5 6h control, 6 6h promoter and 4 6h enhancer. All qRT-PCR experiments were performed using ViiA7 System (Applied Biosystem).

## HEK293 cell culture

HEK293/hTLR4A-MD2-CD14 Cells (Invivogen) were selected for functional follow up of the *IRF1 trans*-eQTL because of the ease of transfection of HEK293 cells and the addition of the TLR4 receptor, which is essential for cellular response to LPS. Cells were cultured in DMEM supplemented with 4.5 g/l glucose (Corning), 10% fetal bovine serum (Sigma-Aldrich), 1% penicillin/ streptomycin (Corning) and 1% L-glutaMAX (gibco). Cells were passaged using cell scraping to avoid damaging the cell surface receptors.

## LPS concentration and time point optimization

To determine the optimal LPS concentration and harvesting timepoints for the HEK293-TLR4 cells, we tested their transcriptomic response to LPS under multiple conditions. HEK293-TLR4 cells were plated into three wells in 24-well plates with 180,000 cells per well. 24 hours later, 0 ng, 500 ng, 1000 ng or 2000 ng of LPS (Invivogen) per mL was added to the cells. RNA was extracted at three timepoints for the 500 and 2000 ng conditions (90m, 6h, 24h) and six timepoints for the 1000 ng condition (45m, 90m, 3h, 6h, 12h, 24h), by adding 500

uL of IBI Isolate DNA/RNA Reagent (IBI Scientific) directly to cells on the plate and stored at -80C until extraction.

## RNA extraction and RNA-seq library preparation

RNA was extracted following the Direct-zol RNA MicroPrep kit (Zymo Research) manufacturer's instructions. RNA was treated with DNAse I (Ambion) and enriched for mRNA using Dynabeads mRNA DIRECT Purification Kit (Thermo Fisher). cDNA was generated using a custom scaled-down modification of the SMART-seq protocol [36]. cDNA was synthesized from RNA input using Maxima H Minus Reverse Transcriptase (Thermo Fisher Scientific). It was then amplified using Kapa HiFi 2X Ready Mix (Kapa Biosystems) and cleaned using 0.9X Ampure beads (Beckman Coulter). Finally, cleaned cDNA samples were tagmented and indexed using the Nextera XT DNA Library Prep Kit (Illumina). Library size and tagmentation were confirmed using the TapeStation HS D1000 kit (Agilent). Libraries were pooled in equal molarity and sequenced with the NextSeq 550 High-Output kit (Illumina) with paired-end 75 bp reads.

## RNA-seq data processing

Reads were first trimmed of adapters using trimmomatic, then aligned to the hg19 genome using STAR 2-pass mapping. Gene counts were calculated with FeatureCounts using Gencode v19 gene annotations.

## CRISPRi of IRF1 promoter and enhancer locus

In order to determine whether the region of the eQTL variant regulates IRF1, and the effect of IRF1 perturbation in our cell line, we performed CRISPRi experiments targeting both the variant locus and the IRF1 promoter. HEK293-TLR4 cells were plated in three 24-well plates with 120,000 cells/well in 1 ml of DMEM. 24 hours later, the medium was replaced with 0.5 ml of OptiMem (Gibco) and cells were transfected with 1.5 ul of lipofectamine 3000 (Thermo Fisher Scientific), 200 ng of CRISPR-KRAB-MeCP2 vector (Addgene 110821) and 50 ng of gRNA gblock (IDT) (4 wells each received a gRNA targeting EGFP as a neutral control ('GGTG GTGCAGATGAACTTCA'), 14 bp downstream of the IRF1 promoter ('GTCTTGCCTCGAC-TAAGGAG') or the enhancer ('TTCTCTGTAGCCCTTGTATT')). After 28 hours, 1 ug/mL of LPS (Invivogen) was added to the 90 m and 12 h samples and nothing to the control samples, with four biological replicates of each gRNA for each LPS treatment (36 total samples). Cells were collected at their respective time point with TRIzol reagent (Thermo Fisher Scientific) and stored at -80C before RNA extraction and RNA-sequencing.

## Genome editing

In order to validate the *cis* and *trans* reQTL associations of rs17622517, we edited the HEK293-TLR4 cells using CRISPR/Cas9 genome editing. A gRNA ('TTCTCTGTAGC CCTTGTATT') was designed with an NGG PAM and a cut site 1 bp downstream from rs17622517. The gRNA was ordered as a single stranded oligo gblock from IDT and amplified using 2 50 uL reactions of Q5 High Fidelity 2X Master Mix (NEB). Cells were transfected with 0.5 ug gRNA gblock and 2.5 ug px458 plasmid (Addgene plasmid # 48138) containing spCas9 and GFP, using lipofectamine 3000 (Thermo Fisher Scientific). 24-hours later, cells then underwent fluorescence-activated cell sorting for GFP+ cells using a Sony SH800Z cell sorter to enrich for transfected cells. Efficiency of editing was tested using a T7E1 assay and

electrophoresis gel to detect presence of NHEJ. The GFP+ cells were also sorted as single cells into 15 96-well plates and expanded into monoclonal cell lines.

Clones were genotyped by creating an amplicon library for each clone from gDNA using nextera primers capturing a 218 bp amplicon containing the variant locus. Indexing PCR was performed using primers specific to the constant sequence on the Nextera primers, resulting in dual-barcoded amplicons with Illumina adapters. Libraries were mixed in equal volume and sequenced on the MiSeq using 150 bp paired-end reads. Fastq files generated by the Illumina software were trimmed for adapter sequences and quality using trimmomatic. Reads were aligned to the genomic locus and categorized as no edit or NHEJ (with indels) using Edityper (Yahi et al. in prep). Since the cell line is triploid at this genomic locus, clones were considered heterozygous if they had NHEJ rate between 20–70%. Clones with less than 10% NHEJ were considered wild type and clones with greater than 90% NHEJ were considered homozygous edited. Five each of wild type, heterozygous and homozygous edited were selected for follow up. BAM files from the selected clones were visually inspected to confirm genotype. In addition, a 3,496 bp PCR followed by electrophoresis gel was performed on the clones in order to check for larger indels not captured by the shorter amplicon libraries.

## LPS treatment of edited clones

To detect the reQTL effect of the edited locus, we performed LPS treatment followed by RNA-sequencing on each isolated edited and wild type cell line. Each clonal cell line was plated into three wells (one well each for control, 90m and 12h treatments) in 24-well plates with 180,000 cells per well. 24 hours later, 1ug/mL of LPS (Invivogen) was added to the 90 m and 12 h samples. RNA was extracted at the designated time point by adding 500 uL of IBI Isolate DNA/RNA Reagent (IBI Scientific) directly to cells on the plate and stored at -80C until extraction and RNA-sequencing. This procedure was done twice to reduce potential technical variability from the LPS treatment and sequencing, and after verifying that the results were generally consistent, the gene count matrices from the two runs were summed together for analysis.

## Differential expression analysis

Differential expression analyses for the CRISPRi and CRISPR data were performed using the R package DEseq2 [37]. We first transformed and normalized the gene count matrices with vst(), and did principal component analysis. Differential expression was performed using interaction models. For CRISPRi we used ~gRNA + condition + gRNA:condition where gRNA denotes promoter, enhancer or control gRNA, and condition denotes 0, 90 min or 12 h LPS treatment. Additionally, we looked at the promoter vs control and enhancer vs control gRNA effect within 90m LPS samples and 12h LPS samples, and the effects of the LPS treatments within the control samples. For CRISPR, we used a nested interaction model (expression ~ genotype + genotype:clone + genotype:condition) accounting for the same clone undergoing different LPS treatments. Additionally, we looked at the effects of the two LPS treatments within the WT clones. Prior to p-value correction, genes were discarded if they were not annotated as protein coding or lncRNA or they did not have an average expression of greater than 5 read counts across samples. P-values for differential expression were corrected using Benjamini-Hochberg correction, using FDR<0.05 threshold.

Enrichment analysis of significantly differentially expressed genes was done using DAVID biological process gene ontology enrichment [38], using Benjamini-Hochberg corrected p-values and FDR<0.05 except for comparison between edited clone and CRISPRi differential expression, where FDR <0.25 was used.

## Supporting information

**S1 Table. LPS 6h Trans-eQTLs.** Significant LPS 90 min cis-reQTLs in monocytes were tested against all expressed probes and filtered for an FDR cutoff of 0.5.
(XLS)

**S2 Table. IRF1 90m cis-eQTL.** Association between genotype of variants within 100 kb of IRF1 transcription start site and IRF1 expression in monocytes treated with LPS for 90 min.
(XLS)

**S3 Table. IRF1 RT-qPCR in THP1 cells.** Relative quantification of IRF1 in RNA extracted from CRISPRi THP1 cells with promoter, enhancer or control gRNAs with and without LPS stimulation.
(XLSX)

**S4 Table. Differential gene expression (FDR 0.05) for CRISPRi samples.** hTLR4 Cells were transfected with control, enhancer and promoter gRNAs and a CRISPRi construct. They were then treated with no LPS, LPS for 90 min and LPS for 12h. Differential expression analysis was performed for LPS treatment in CRISPRi control samples, gRNA condition in different LPS treatments, and the interaction between LPS treatment and gRNA condition.
(XLS)

**S5 Table. Gene ontology enrichment for CRISPRi differentially expressed genes.** Gene ontology enrichment analysis was performed on gene sets from CRISPRi differential expression analysis.
(XLS)

**S6 Table. Differential gene expression (FDR 0.05) for CRISPR edited clones.** hTLR4 Cells were transfected a CRISPR/Cas9 construct and a gRNA specific for the rs17622517 locus. Wild type, heterozygous or homozygous clones were isolated and then treated with no LPS, LPS for 90 min and LPS for 12h. Differential expression analysis was performed for LPS treatment in WT clones, and the interaction between genotype and LPS treatment.
(XLS)

**S7 Table. Gene ontology enrichment for CRISPR edited differentially expressed genes.** Gene ontology enrichment analysis was performed on gene sets from CRISPR edited differential expression analysis.
(XLS)

**S1 Fig. Trans-eQTL analysis with 126 cis-eQTL.** Histogram of the number of trans probes associated to each cis-eQTL SNP in monocyte trans-eQTL study at 0.05 FDR (A) and number of probes associated with each SNP at 0.5 FDR (B).
(TIFF)

**S2 Fig.** rs17622517 cis association with expression of genes within 1 MB in monocytes in control (A) and LPS 90 min stimulation (B).
(TIFF)

**S3 Fig. Trans-eQTL associations for rs17622517.** Twelve most significant trans-eQTLs associated with rs17622517 under 6h LPS at FDR < 0.05.
(TIFF)

**S4 Fig. Secondary eQTL signal in the IRF1 locus.** A) Locus zoom plot highlighting LD pattern for second top SNP in IRF1 cis-eQTL and B) cis-eQTL effect of rs147386065 on IRF1, C) trans-eQTL effect sizes of the significant rs17622517 trans-eQTLs (FDR < 50%) versus effect

sizes of rs147386065 trans effects on the same genes.
(TIFF)

**S5 Fig. Comparison of LPS response in HEK293-hTLR4 cells versus monocytes.** A) expression of IRF1 for the different conditions; B-C) number of rs17622517 trans-eGenes that have their maximum expression at each time point in primary monocytes using data from Kim-Hellmuth et al. 2017 (B), and in HEK293-TLR4 cells (C). The peak trans-eGene expression is at 6 hours in monocytes and at 12 hours in HEK293-TLR4 cells. D) Overlap of genes expressed in monocytes and HEK-hTLR4 cells. Monocyte expression was quantified with microarray (in Kim-Hellmuth, 2017) where expressed probes were defined as having a p value of detection less than 0.0001. HEK293-hTLR4 cells were quantified with RNA-seq and expressed genes were defined as having an average expression across samples of at least 5 reads. E) Correlation of the log2 fold change (log2FC) in HEK-hTLR4 versus monocytes for the top 100 differentially expressed genes (by absolute value of fold change) in HEK-hTLR4 cells. F) Correlation of log2FC in HEK-hTLR4 versus monocytes for the top 100 differentially expressed genes in monocytes.
(TIFF)

**S6 Fig. Western blot confirming dKRAB-dCas9-MECP2 protein in THP1 cells.** Western blot with anti HA-tag of THP1 WT, THP1 KRAB-dCas9-MeCP2 polyclonal and monoclonal cell lines. Mouse GAPDH was used as loading control.
(TIFF)

**S7 Fig. Effect of LPS stimulation on IRF1 in monocytes.** Expression change in IRF1 for 134 donors from Kim-Hellmuth, 2017), showing a mean fold change of 1.23 between control and LPS 90m samples (Wilcoxon $p < 2.2^{-16}$).
(TIFF)

**S8 Fig. Enhancer silencing effects on gene expression in cis.** Expression levels of the genes within 1Mb in HEK293-TLR4 cells with and without CRISPRi silencing of the rs17622517 locus. The lack of significant difference indicates that this putative enhancer does not have a strong effect on other genes in cis.
(TIFF)

**S9 Fig. Enhancer and promoter silencing effects for different LPS time points.** Number of significantly differentially expressed genes in promoter (A) and enhancer (B) versus controls at the respective LPS condition.
(TIFF)

**S10 Fig. Genotyping of edited clones.** A) Electrophoresis of long-range PCR products showed a large deletion in one clone (green) with that was excluded from further analysis; B) genotypes of clones selected for analysis.
(TIFF)

**S11 Fig. Principal component analysis of the two replicate CRISPR experiments.** A) Principal components 1 and 2 for all clones across both experiments, highlighting condition and experiment. B) Principal components 3 and 4 for all clones under control conditions, highlighting clone identity.
(TIFF)

**S12 Fig. Correlation of differential expression between CRISPR and CRISPRi.** Correlation of gene expression fold changes at 12h LPS condition for the top 100 differentially expressed genes (by fold change) in wild type versus heterozygous clones (A & C) and wild type versus

homozygous clones (B & D) versus fold change for promoter (A & B) and enhancer (C & D) CRISPRi silencing. Correlation of 0.05 FDR differentially expressed genes in heterozygous versus homozygous clones versus fold change for promoter (E) and enhancer (F) CRISPRi silencing.
(TIFF)

**S13 Fig. Correlation of trans-eQTL effect sizes and experimental perturbations.** Correlation of 0.5 FDR trans-eQTL effect size (allelic fold change) with differential expression in (A) wild-type versus CRISPR-edited homozygous clones, (B) CRISPR-edited heterozygous versus homozygous clones and (C) CRISPRi promoter silencing versus control samples. D) Correlation of 0.05 FDR trans-eQTL effect size with differential expression in CRISPR edited heterozygous clones.
(TIFF)

## Acknowledgments

We thank Harm Wessel and Neville Sanjana for assistance with the THP1 cell line.

## Author Contributions

**Conceptualization:** Margot Brandt, Tuuli Lappalainen.

**Formal analysis:** Margot Brandt, Sarah Kim-Hellmuth, Marcello Ziosi, Tuuli Lappalainen.

**Funding acquisition:** Tuuli Lappalainen.

**Investigation:** Margot Brandt, Sarah Kim-Hellmuth, Marcello Ziosi, Alper Gokden, Aaron Wolman, Nora Lam, Yocelyn Recinos.

**Methodology:** Margot Brandt, Sarah Kim-Hellmuth, Marcello Ziosi.

**Project administration:** Tuuli Lappalainen.

**Resources:** Sarah Kim-Hellmuth, Marcello Ziosi, Zharko Daniloski, John A. Morris, Veit Hornung, Johannes Schumacher.

**Supervision:** Tuuli Lappalainen.

**Visualization:** Margot Brandt, Tuuli Lappalainen.

**Writing – original draft:** Margot Brandt, Tuuli Lappalainen.

**Writing – review & editing:** Margot Brandt, Tuuli Lappalainen.

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
