## [Decision Letter · Decision Letter 0]

5 Jun 2020

Dear Dr Lappalainen,

Thank you very much for submitting your Research Article entitled 'An autoimmune disease risk variant has a trans master regulatory effect mediated by IRF1 under immune stimulation' to PLOS Genetics. Your manuscript was fully evaluated at the editorial level and by independent peer reviewers. The reviewers appreciated the attention to an important problem, but raised some substantial concerns about the current manuscript. In particular, both reviewers raised the concern that the CRISPRi and CRISPR experimental results do not support the causal role of the SNP and the mediation by IRF1. Based on the reviews, we will not be able to accept this version of the manuscript, but we would be willing to review again a much-revised version. We cannot, of course, promise publication at that time.

If you decide to revise the manuscript for further consideration at PLOS Genetics, please aim to resubmit within the next 60 days, unless it will take extra time to address the concerns of the reviewers, in which case we would appreciate an expected resubmission date by email to plosgenetics@plos.org.

[LINK]

We are sorry that we cannot be more positive about your manuscript at this stage. Please do not hesitate to contact us if you have any concerns or questions.

Yours sincerely,

Xin He

Guest Editor

PLOS Genetics

Scott Williams

Section Editor: Natural Variation

PLOS Genetics

Reviewer's Responses to Questions

**Comments to the Authors:**

Reviewer #1: The authors map trans eQTLs in monocytes from 134 donors stimulated with LPS for 6h, restricting the analysis to SNPs previously identified as response cis-eQTLs at an earlier timepoint (90 mins LPS treatment). They identify a locus that is a cis-eQTL for IRF1 and is associated with a large number of IRF1 target genes in trans, as well as being a GWAS locus. They aim to validate this putative enhancer using CRISPRi and CRISPR in a modified HEK293 cell line, and present a model of how the risk allele impacts gene expression.

It is important to conduct these functional analyses to follow up eQTL studies and I appreciate the challenges with being able to design the perfect experiment to directly address the difficult problem of identifying causal SNPs and genes. However, my major concern is that the results do not support the strong conclusions made in the manuscript. The results presented are only suggestive of a role of IRF1 and rs17622517 in the nearby enhancer locus and additional work will be required to confirm they are the causal gene and SNP.

Major comments:

1. The results of the CRISPRi experiment do not support the strong conclusion that IRF1 is downregulated by the enhancer silencing

a. As stated on page 10 “The effect of the enhancer silencing is not significantly different from the controls for IRF1”. Given that downregulation of IRF1 was not observed I would suggest that the later sentence “The differential expression of promoter and enhancer have a strong positive correlation...which indicates that enhancer silencing downregulates IRF1” is too strong a conclusion and should be rephrased.

b. On page 10: “the response being similar to the IRF1 promoter silencing further provides strong evidence that it is indeed an enhancer of IRF1”. This evidence is suggestive rather than strong.

c. Figure 2c: Is the upregulation of IRF1 expression in response to LPS less than would be expected based on the previous microarray study and the response in the CRISPR experiment? While statistically significant, from the figure it looks like a very modest upregulation in expression.

d. On page 14 the authors state that “CRISPRi is suboptimal for recapitulating eQTL effects driven by genetic variants”. Could they expand on the motivation for using this technique for this eQTL validation experiment? Have the authors considered if any other experiments could be more informative for determining if the rs17622517 locus is an enhancer for IRF1 e.g. chromatin conformation capture?

e. As described in the points above, the following sentence in the abstract referring to the function as an IRF1 enhancer feels too strong given the results of the CRISPRi silencing experiment “Using CRISPRi silencing, we showed that the SNP locus indeed functions as an IRF1 enhancer with widespread transcriptional effects.”

2. On page 10, could the authors expand upon why they hypothesise that they are “underpowered to detect an effect on IRF1 in cis, but might capture its trans effects” given there is usually better power for the detection of cis effects?

3. The results of the CRISPR experiment do not support the conclusion that rs17622517 is the causal SNP driving the cis and trans eQTL effects. The experiment conducted provides additional supportive evidence for the role of that locus in the regulation of IRF1 expression, but it still doesn’t address the hypothesis that rs17622517 is the causal SNP.

a. Could the authors elaborate on what the correct interpretation of NHEJ is in the context of an eQTL? How should the reader interpret the deletion of the eQTL allele rather than a substitution?

b. On page 12 it is very hard to see the “potential slight genotype effect” particularly in fig 3a.

c. Although the correlations between CRISPR and CRISPRi are statistically significant, they are very weak and what looks like nearly half of the genes show opposite effects between the two experiments.

d. As described in the points above, I do not agree that the results presented support the sentence on page 13 “Having provided experimental evidence that rs17622517 is the causal eQTL variant”

e. Fig 4a shows a weak correlation between the trans-eQTL effect size and the CRISPR experiment which is being driven by a small number of genes. Given that such a relaxed FDR threshold of 0.5 has been used, could the authors comment on whether the genes that do show a good correlation between the experiments are those with the more significant trans eQTL FDR?

f. As described in the points above, the results presented do not support the statement in the abstract “CRISPR further indicated that rs17622517 is indeed a causal variant in this locus, and recapitulated the LPS-specific trans-eQTL signal”

Minor comments:

1. Could the authors clarify if the “strongest trans-eGenes” referred to on page 7 are those with the most significant eGenes rather than necessarily those with the greatest effect size?

2. Page 7: how were the tagging variants identified? Could this SNP be tagging a structural variant?

3. Beyond highlighting that the rs17622517 has a substantial number of trans-eGenes, Figure 1a is initially hard to interpret further. It may be more helpful to plot a histogram of how many cis SNPs have 1 trans-eGene, 2 trans-eGenes etc to highlight the rs17622517 locus but also give a better idea of how many eGenes are detected for the other SNPs.

4. It would be helpful to include the p value and effect size of the cis eQTLs in figure 1b.

5. What does the shading represent on figure 1e?

6. For supplementary figure 5 could the authors clarify in the figure legend what comparisons have been made for the differential expression due to enhancer and promoter silencing in? Is this compared to the same time point in controls?

7. Are the authors concerned that the differential expression lists aren’t enriched for IRF1 targets? Is it something else driving the difference in expression e.g. off target CRISPR effects, or is this related to the time points chosen?

8. Fig 4c is cited in the text before fig 4a. Fig 4d is cited in the text before fig 4b.

Reviewer #2: In « An autoimmune disease risk variant has a trans master regulatory effect mediated by IRF1 under immune stimulation », authors Margot Brandt et al. explore the trans-effect of genetic variation on the transcriptional response of primary monocyte upon LPS stimulation using a data set published previously in Kim-Hellmuth et al. (2017). They identified a risk variant for autoimmune disease, rs17622517, as a master regulator of the innate immune response upon late LPS stimulus through the regulation of IRF1 gene expression. To validate the role of this particular variant in such regulation, they performed CRISPR-based experiments in order to provide functional evidence of its causative effect.

Overall, the results presented by the authors are of interest since only few studies have attempted to develop an in vitro CRISPR-based approach in cellular models to characterize the functional effect of immune-responsive regulatory variants. However, I have major concerns about the conclusions highlighted by the authors.

The experimental work presented here suggest a trans-effect of the rs17622517 locus regulating immune response genes following LPS stimulation, but it is not sufficient to conclude that the effect is modulated by IRF1. Indeed, the authors did not show any direct evidence that the rs17622517 locus is an enhancer of IRF1 expression since the inhibition of this locus by CRISPRi has no effect on IRF1 expression (Wilcoxon p=O.8), as opposed to the inhibition of IRF1 promoter which exhibited a significant down-regulation of IRF1 (Wilcoxon p=7.5x10-7). The same remark can be made for the CRISPR/Cas9 genome editing experiments where the different deletions introduced in the rs17622517 locus have again no effect on IRF1 expression. In addition, the lack of power that they highlighted several times in the article, as well as the relaxed FDR thresholds that they use for the analysis indicate some weakness of the study and highlight the need to be replicated more deeply in order to have confident results. For all these reasons, the authors should be more cautious when claiming that the trans-effect of the rs17622517 locus is modulated by IRF1. The conclusions, as well as the title of this article, should be modified accordingly.

Moreover, the cellular model of HEK293-TLR4 cell line used here does not seem to be well suited to study IRF1 pathway. Although this cellular system is responsive to LPS stimulus, there is no enrichment of IRF1 target genes among differentially expressed genes upon stimulation, as opposed to the results obtained in the primary monocytes. A comparison of the transcriptional response upon stimulation between monocytes and HEK293-TLR4 cell line would be informative (Venn diagram to see the proportion of genes expressed in both cell-types, correlation of the transcriptional response for genes commonly expressed, GO enrichment for these genes…). More particularly, are the 12 trans-eGenes the most strongly associated at FDR=0.05 in monocytes expressed in HEK293 – TLR4 cells? If so, is there a correlation in their expression?

The CRISPR/Cas9 gene editing design used by the authors did not allow the discrimination of the trans-effect at the nucleotide resolution since they introduced deletions surrounding the rs17622517 locus and did not perform SNP editing. They delimited a DNA sequence where a regulatory element may be present and any nucleotide modification within this sequence, including rs17622517, may have a trans-effect. A comment on this issue in the article would be appreciable. In addition, the results regarding the analysis of Het and Hom clones are not convincing because there is no down-regulation of IRF1 expression depending on the genotype classes, as expected. The trans-effect with differential expression in Het and Hom clones showed also an opposing direction of the correlations, which is counter-intuitive, and the variability within genotype classes seems very high. The authors used ranges of NHEJ rate to determine the different genotype classes (NHEJ<10% for WT, NHEJ=20-70% for Het and NHEJ>90% for Hom). Did they choose these ranges arbitrary or this decision was supported by biological/statistical arguments? Moreover, how did the authors estimate the number of clones to test per genotype class? What was the transcriptional variation among clones within each class? Given the moderate effect size of rs17622517 locus and knowing the transcriptional variability among clones within each class, they should calculate the statistical power to detect a biological effect using 13 clones.

My general comment on this article is that the interpretation of the data should be more rigorous:

Depending on the biological hypothesis that the authors want to highlight, they considered different FDR thresholds, from 5% FDR up to 50% FDR. They should explain on which criteria they selected these thresholds and what are the consequences in terms of data significance. It reveals the clear lack of statistical power in their analyses, which results in a doubt about the robustness of the genetic associations.

In the figure Fig. 1a, the authors reported that the rs17622517 locus presented “substantially” a larger number of trans-eGenes with respect to other loci at a FDR=0.5. First they should compare groups using a FDR of 0.05 and second test if the difference is statistically significant. In general, the authors should avoid the terms “substantially” or “suggestive” when no statistical support is provided.

The authors mention that the lead variant rs17622517 had no LD-tagging variants but no information on the processing of the genotyping data has been reported in the article. What is the total number of SNPs considered in the analyses? Has SNP imputation been performed? Has the genetic sub-structure of the population been accounted for in the eQTL analysis? Has a fine mapping of the region been performed to make sure that rs17622517 was the best SNP associated with the disease, as well as with gene expression phenotypes?

Several replicates have been performed in this study but no analysis of the replicates have been reported by the authors in the article, knowing that such analysis is essential for the validation of the experimental settings. Please provide measures of technical variability and show that the biological effect that you expect is higher. Regarding the two replicates that were done for each of the 13 clones, the reads from the replicates were combined. Did the authors quantify the technical noise between replicates? Did they perform some quality checks to ensure that the sequencing data from the two replicates could be merged (correlations, variation between replicates…).

Regarding in the analyses ran in the Figure 3 d-e, the authors used the set of genes that were significantly differentially expressed in heterozygous lines compared to homozygote lines because the effect of the genetic mutations was best captured with respect to the other genotype comparisons. Please provide the biological sense of comparing the fold change of these genes with the fold change obtained in the CRISPRi promoter vs control or in the CRISPRi enhancer vs control. Such comparisons should be performed with the list of genes captured in Het vs WT or Hom vs WT, instead of Het vs Hom.

Minor comments:

When multiple expression probes measure the same gene, what was the correlation between probes for the trans-eGenes?

Is the background gene set used in the enrichment analysis for IRF1 target genes based on a pool of genes expressed in all conditions or only on gene expressed in the condition of interest?

Was the expression data corrected for technical variability?

Please add in the supplementary tables 3 and 5 the name of the genes using HUGO nomenclature.

In Supplementary Figure 6B, the orange line for gDNA should be replaced by the genomic sequence of the region.

Regarding the clones edited by CRISPR/Cas9, it would be appreciable to have their ID on the corresponding figures.

**Have all data underlying the figures and results presented in the manuscript been provided?**

Reviewer #1: Yes

Reviewer #2: Yes

PLOS authors have the option to publish the peer review history of their article (what does this mean?). If published, this will include your full peer review and any attached files.

Reviewer #1: No

Reviewer #2: Yes: Helene Quach

---

## [Decision Letter · Decision Letter 1]

28 Apr 2021

Dear Dr Lappalainen,

Thank you very much for submitting your Research Article entitled 'An autoimmune disease risk variant has a trans master regulatory effect mediated by IRF1 under immune stimulation' to PLOS Genetics.

The manuscript was fully evaluated at the editorial level and by independent peer reviewers. The reviewers appreciated the attention to an important problem, but raised some substantial concerns about the current manuscript. Based on the reviews, we will not be able to accept this version of the manuscript, but we would be willing to review a much-revised version. We cannot, of course, promise publication at that time.

Should you decide to revise the manuscript for further consideration here, your revisions should address the specific points made by each reviewer and the editorial comments below our signatures. We will also require a detailed list of your responses to the review comments and a description of the changes you have made in the manuscript.

If you decide to revise the manuscript for further consideration at PLOS Genetics, please aim to resubmit within the next 60 days, unless it will take extra time to address the concerns of the reviewers, in which case we would appreciate an expected resubmission date by email to plosgenetics@plos.org.

[LINK]

We are sorry that we cannot be more positive about your manuscript at this stage. Please do not hesitate to contact us if you have any concerns or questions.

Yours sincerely,

Xin He

Guest Editor

PLOS Genetics

Scott Williams

Section Editor: Natural Variation

PLOS Genetics

Both reviewers acknowledged the extra efforts made by the authors, but raised the concern of the strength of the evidence presented, particularly about the effect of the SNP on IRF1 expression. We would suggest the authors to better address the limitations of their evidence. Additionally, please change the title to: "An autoimmune disease risk variant: a trans master regulatory effect mediated by IRF1 under immune stimulation?"

Reviewer's Responses to Questions

**Comments to the Authors:**

Reviewer #2: In the revised version of “An autoimmune disease risk variant has a trans master regulatory effect mediated by IRF1 under immune stimulation”, the authors Margot Brandt et al. have significantly improved the quality of the manuscript by bringing new experimental data (i.e. generation of THP-1 CRISPRi cell lines and RT-PCR experiment to measure IFR1 gene expression) in order to support the trans-effect of rs17622517 on immune response following LPS activation, via the regulation of IRF1. They have also added supplementary analyses (i.e. comparison of LPS response in HEK293-hTLR4 cells versus monocytes) and technical information that allow a better understanding of the article.

The association between rs17622517 and IRF1 expression remains suggestive, preventing any clear demonstration of a trans-effect of rs17622517 following LPS activation, even in THP-1 cells, as shown in the new figure 2b. Regarding this new figure, enhancer silencing leads to significant upregulation of IRF1 at 0h LPS (t-test p= 0.018). Since we do not expect an effect of the enhancer at baseline on the basis of the eQTL mapping performed in primary monocytes (Figure 1b), please comment on your results. In addition, the authors used a t-test to show a suggestive down-regulation of IRF1 at 90 min LPS (t-test p=0.078), but nothing was specified about the normality of the distribution. According to the low number of samples, I think a Wilcoxon rank-sum test would be more appropriate. I suggest that the authors re-calculate the p-values accordingly. Finally, to appreciate the variation of IRF1 expression among THP-1 clones, the box plot should show the median value with the interquartile range and individual values should be added to the figure.

In general, the authors have made a lot of efforts to discuss the lack of power in their experimental settings preventing them to clearly demonstrate the causal role of rs17622517 in the inter-individual response to LPS stimuli via regulation of IRF1, and have softened their claims accordingly.

Reviewer #3: I am commenting on the revised version of this manuscript, not having given feedback on the original. In general, the authors have made substantial revisions in light of the comments on the original manuscript. They show several lines of evidence which, whilst individually weak, suggest that a variant near IRF1 is associated with IRF1 expression levels in early macrophage LPS response, and with differences of IRF1 targets in trans later in the same response. The CRISPRi data suggest this region, which harbors a variant associated to IBD, is an enhancer, as there are marginal effects on LPS-induced IRF1.

This work is a good cautionary tale in how hard it can be to unravel trans-eQTLs - both temporally and in terms of effect size. The authors have to compile multiple strands of individually incomplete evidence to arrive at their current narrative. I have a couple of things for them to consider:

- There is no co-localization analysis between the IBD GWAS data and at least the IRF1 cis-eQTL data, which might indicate a shared effect. In the title and discussion, the authors imply that this is likely a disease-relevant mechanism, but some acknowledgement of the limited resolution of fine-mapping should be made if no sharing can be shown.

- IRF1 is expressed in a wide variety of immune cell types. The possibility that this mechanism, if relevant to disease, may be acting in some other cell type deserves some acknowledgment.

- There is no discussion or data on whether this trans-eQTL actually changes anything about macrophage response to LPS. If no cellular phenotype can be detected, why would this effect be likely to have a broader physiological effect like influencing disease risk? I would suggest this merits at least some discussion as a limitation on the interpretation of the data presented.

**Have all data underlying the figures and results presented in the manuscript been provided?**

Reviewer #2: None

Reviewer #3: Yes

PLOS authors have the option to publish the peer review history of their article (what does this mean?). If published, this will include your full peer review and any attached files.

Reviewer #2: **Yes: **Helene Quach

Reviewer #3: No

---

## [Decision Letter · Decision Letter 2]

25 Jun 2021

Dear Dr Lappalainen,

We are pleased to inform you that your manuscript entitled "An autoimmune disease risk variant: a trans master regulatory effect mediated by IRF1 under immune stimulation?" has been editorially accepted for publication in PLOS Genetics. Congratulations!

Yours sincerely,

Xin He

Guest Editor

PLOS Genetics

Scott Williams

Section Editor: Natural Variation

PLOS Genetics

Comments from the reviewers (if applicable):

Reviewer's Responses to Questions

**Comments to the Authors:**

Reviewer #2: The authors Margot Brandt et al. took into account all my suggestions in their last version.

**Have all data underlying the figures and results presented in the manuscript been provided?**

Reviewer #2: Yes

PLOS authors have the option to publish the peer review history of their article (what does this mean?). If published, this will include your full peer review and any attached files.

Reviewer #2: **Yes: **Helene Quach

**Data Deposition**

http://datadryad.org/submit?journalID=pgenetics&manu=PGENETICS-D-20-00488R2

**Press Queries**

---

## [Editor Report · Acceptance letter]

22 Jul 2021

PGENETICS-D-20-00488R2 

An autoimmune disease risk variant: a trans master regulatory effect mediated by IRF1 under immune stimulation? 

Dear Dr Lappalainen, 

We are pleased to inform you that your manuscript entitled "An autoimmune disease risk variant: a trans master regulatory effect mediated by IRF1 under immune stimulation?" has been formally accepted for publication in PLOS Genetics! Your manuscript is now with our production department and you will be notified of the publication date in due course.

With kind regards,

Katalin Szabo

PLOS Genetics

On behalf of:
